# Micromechanical Model and Thermal Properties of Dry-Friction Hybrid Polymer Composite Clutch Facings

**DOI:** 10.3390/ma13204508

**Published:** 2020-10-12

**Authors:** Roland Biczó, Gábor Kalácska, Tamás Mankovits

**Affiliations:** 1Faculty of Mechanical Engineering, Szent István University, Páter Károly u. 1., H-2100 Gödöllő, Hungary; kalacska.gabor@gek.szie.hu; 2Department of Mechanical Engineering, Faculty of Engineering, University of Debrecen, Ótemető u. 2-4., H-4028 Debrecen, Hungary; tamas.mankovits@eng.unideb.hu

**Keywords:** dry-friction clutch, fiber-reinforced hybrid composite, stiffness matrix, thermal properties, thermomechanical input

## Abstract

Fiber-reinforced hybrid composites are the most commonly used dry-friction materials in the automotive industry. On the other hand, the great variety of components utilized these days in such material systems often requires identification investigations for a complex characterization. The development history of clutch materials was reviewed, highlighting and understanding the milestones and efforts leading to the creation of these materials. Investigations were performed to determine mechanical stiffness matrix parameters and thermal properties of a woven fiber yarn (glass fiber with aromatic polyamide, copper, and poly-acrylic-nitrile (PAN) reinforced friction material, revealing and solving challenges faced during identification efforts. This study grants an effective reference and a novel guidance for material identification methods for similar complex materials, and the results provide input parameters for thermomechanical simulation contact model development, which will cover friction material lifetime effects on dry clutch tribology in a future study.

## 1. Introduction

Clutches are responsible for transferring the kinetic energy of a rotating crankshaft coupled to a power source toward a transmission system and wheels in automotive applications. Figure 1 illustrates a dry, single-disc type of clutch with a fiber-reinforced friction facing. Heat generated via slippage is absorbed then dissipated to the atmosphere by the clutch system [1].

The operating characteristic requirements of composite friction materials of automotive clutch systems are responsible for determining the remarkable structural and material development of the field through the last century.

Milestones of this development are summed up in Table 1, leading to the modern hybrid polymer composite friction material clutches of the present day.

During the last century, the development of dry sliding clutch friction facing materials was mainly determined by the construction of the clutch. Newer types (such as belt drives, conical, and single-disc clutches) then surfaced, as shown in Table 1. Consequently, new achievements in material technology led the geometrical and dimensional restriction-led development of clutch structures toward an optimum design. During the second half of the century, development trends compared to similarly loaded friction applications from different industrial fields were led by common discoveries of the scientific and industrial world of tribology. For instance, material development was highly affected by the efforts to replace asbestos. In the meantime, discovering flexible resins made fiber-reinforced materials widespread [2]. However, parallel construction and material development allowed improving lubrication, with the utilization of newly discovered components for friction materials and certain effective machining methods [3]. As Table 1 illustrates, the synergy and fusion of these two development paths created today’s modern, compact clutches that are resistant to a great variety of extreme challenges. Organic friction facing materials are composites, manufactured via the scatter wound process. The fibers are impregnated with a matrix, then preformed into a wreath, cured, and ground to a final geometry [4,5]. The orientation of fiber reinforcement was determined by the special direction of the friction force impact to create resistance against it. It also influenced the manufacturing method. The different manufacturing stage results of these products are illustrated in Figure 1b.

Requirements derived from the loads and environment of polymer composite friction materials for clutches of present day not only determined their deformability and the stability of their friction coefficient and wear resistance with heat resistance against thermal loads, but also influenced their strength against transient mechanical impacts or high rotational velocity. Components of these materials are usually classified as reinforcements, binders, friction modifiers, and fillers. On the other hand, this separation mainly has significance during production. The reason for this is that a component from one group usually affects the behavior of the material from the aspect of another component group. For example, fibers (typically aromatic polyamide, glass, carbon, steel, cellulosic fiber, thermoplastic fiber, and asbestos, which is banned) play a critical role in determining not only the mechanical strength, but also the thermal resistance and friction/wear properties of these materials [6]. For instance, Bijwe [7] stated that, among those fibers responsible for high strength and modulus, aromatic polyamide guarantees high thermal stability (e.g., carbon and steel fibers), good wear properties, and a stable coefficient of friction.

Friction material field investigations usually detail one main topic from the mechanical, thermal, and tribological ternary while reflecting on the requirements, loads, and operating conditions, even though separating the different loads occurring in a clutch disc facing may result in difficulties due to their complexity. The main topics include mechanical investigations, thermal aspects, or tribological researches [2].

From a mechanical aspect, parameters affecting warpage, shrinkage, and residual stress, the adhesion between fiber and matrix and its governing parameters, fiber–fiber friction conditions, and the influence of waving technology parameters are investigated. In addition, different types of mechanical tests are usually performed [2]. During the creation of hybrid composites, achieving the required geometry can be a fundamental challenge; thus, Zarrelli et al. [8] concluded that selecting manufacturing parameters wisely reduces cost and prevents problems such as warpage, residual stress, micro cracks, and fiber–matrix failures.

The typical curing method of thermoset polymers consists of two steps: an isotherm step and cooling. During the former, the polymer suffers shrinkage as it solidifies from a liquid state while the fiber remains unchanged. During cooling, the fiber shrinks as well, but with a different thermal expansion coefficient to the matrix. In this way, the residual stress during curing is governed by volume change and material properties [9]. Gurunath et al. [10] developed a new resin for friction composites with phenolic resin binder in order to avoid shrinkage (an end production issue). Zarrelli et al. [8] also investigated manufacturing failures such as spring-in phenomena, warpage, and residual stresses (due to curing-induced chemical shrinkage, heat expansion, and change in viscoelastic modulus) with a novel resin sensitivity model, while trying to reduce residual thermal stress caused by processes in carbon fiber thermoplastics. Greisel et al. [11] pointed out that an increase in interfacial fracture toughness can be achieved by annealing the untreated composite at temperatures beyond the glass transition temperature of the matrix. The control of curing temperatures for minimizing shrinkage and the separation on different stresses caused by this phenomena and thermal expansion were the goals of Kravchenko et al. [12] when creating a multiscale model for the response of a bimaterial “thermostat”.

Fiber–matrix adhesion and fiber–fiber friction are also influential topics explored in the mechanical field. One of the disadvantages of fiber reinforcement with natural fibers is the weak adhesion between the fiber and matrix, which can have a negative effect on the physical and mechanical properties due to the incompatibility of the hydrophilic surface of the fiber and the nonpolar surface of the polymer [13]. Regarding inner friction, de Lange et al. [14] found evidence by conducting pull-out tests that dispersive and polar interactions created through curing can increase the surface energy and strengthen adhesion between aromatic polyamide fibers and different matrices. Xie et al. [13] discovered that silane coupling agents not only increase the interfacial adhesion of natural fibers to the target polymer matrices, but also improve the mechanical and outdoor performance of the whole composite.

From a manufacturing aspect, Bigaud et al. [15] stated that, using a special so-called three-dimensional (3D) braiding technique, carbon fiber woven with PA12 at different braiding angles resulted in different stiffness and strength values. Only some studies covered the effect of waving parameters since these are well written in patents, while the woven wreath must reach a certain density, which makes these manufacturing set-up parameters almost unchangeable. The sensitivity of the mechanical properties of the clutch facing to waving parameters is an important topic to create optimal geometries.

Basic mechanical tests are essential to determine the mechanical properties of a material. However, only a few articles dealt exclusively with dry clutch facing materials. These tests are not only essential for new materials but are also part of manufacturing in the testing phase. Nevertheless, a numerical model is more effective to cut testing costs resulting from expensive lifecycle investigations. By proposing three types of failure criteria in three dimensions (maximum stress; Tsai–Hill, Tsai–Wu; Hashin–Rotem, Sun, NU) Daniel et al. [16] developed test methods for complete mechanical characterization of textile composites.

Among **thermal** topics, most articles aimed to investigate hotspots (small areas where extremely high temperatures and pressures occur when speed is higher than a critical value [2]), thermo-elastic instability, curing induced shrinkage, thermal properties affected by material or manufacturing parameters, and third body phenomena. Due to complex load cases, the goal with thermal models is simplification in order to reduce computational time and costs. Today’s finite element software contains special features making thermal numerical analysis possible.

The mechanism of hotspots consists of an instability that is part of the sliding system loaded by friction heat in accordance with the pv (MPa·m/s) theory of polymer sliding surfaces. The heat flux generated on the surface of the contact is proportional to the contact pressure, where higher temperature increases in higher-pressure areas lead to increased thermal expansion, thereby further increasing the local pressure. This phenomenon, which makes the pressure distribution inhomogeneous, is called frictionally induced thermo-elastic instability (TEI). Zagrodzki et al. [17] simulated unstable thermoelastic friction system behavior (due to material composition and geometry failures) in a wet multidisc clutch, finding that thermoelastic stability can be improved (considering manufacturing limitations) with the reduction of steel disc thickness and hotspots can be controlled by the modulus of elasticity. The change in hotspot location after cooling in a frictional sliding system was presented by Ahn and Jang [18] via a thermo-elastoplastic instability (TEPI) involving a two-dimensional contact problem-solving transient finite element simulation.

Among the manufacturing steps, curing may be the most influential and the most unpredictable phase to determine the final shape due to shrinkage affecting thermal and surface properties. The volume shrinkage is present as chemically induced shrinkage in the resin of friction composites, which clearly affects the residual stresses and the warpage phenomena [19]. Coupling between volume variation and thermal gradients was considered by Nawab et al. [20] for modeling residual strains and stresses to investigate chemical shrinkage of thermoset polymers. They concluded that, since for an equal mass the chemical shrinkage of fibre carrying resin is less than that of neat resin, composite material residual stresses resulting from shrinkage are hindered by fibers.

Higher fiber-to-matrix ratios can rapidly evacuate the frictionally induced heat when cyclic engagements are repeated—a phenomenon not present with a single engagement, as discovered by Khamlichi et al. [4].

An energy storage and conduction-capable uniform layer represented the third body in the thermal numerical model of Majcherczak et al. [21].

On the basis of loads operating under hybrid friction composites, their behavior can be tabulated as impact and fatigue response. Sfarni et al. [5] found connection between these two types. They pointed out that imbedding—a phenomenon developed during the lifetime of a friction clutch due to steel parts causing residual deformation of the friction material—is partially induced by high mechanical and thermal impact loads and intensive wear due to local contact pressure distribution peaks. On the other hand, Menday and Rahnejat [22] found evidence that fatigue response can influence impact the behavior of dry-friction clutch facings. They illustrated that judder—an unwanted axial vibration occurring as an impact load—is less likely to harm the transmission system, if the gradient of the alteration of the coefficient of friction of the friction facing during lifetime remains positive or at least 0.

Tribological scholars often detail the effects of components and manufacturing parameters on frictional properties of the material. Friction coefficient and wear are usually observed and evaluated via pin-on-disc test set-ups. Tribological behavior, along with harmful effects of the frictional contact, is often described in empirical or physical equation-based models. However, our aim here is to establish a coupled thermomechanical model of dry-friction hybrid composite clutch facings, whereas tribological aspects will be covered in future studies by expanding the modeling capabilities of the current concept detailed in this study.

To conclude, clutch composite friction materials developed in strong relation to clutch construction through material industry milestones, such as the discovery and utilization of flexible resins, developments regarding different reinforcing fibers, and environmental and healthcare aspects, resulted in today’s state of-the-art applications [2]. Open questions of the field still circulate around friction behavior under different conditions, along with wear characteristic sensitivity, thermal loads and responses, and even manufacturing steps, with studies performed to determine warpage, shrinkage, and residual stresses, the effects of different fillers, production-induced problems, or biodegradable materials [3].

Usually it is difficult to separate the thermal, tribological, and mechanical effect of material components and manufacturing parameters. The complex synergy of these systems determines the properties. Moreover, properties of the components have no straight effect on final values since the cross-effects take a main role without considering the behavior under loads.

Although there are many considerable research efforts to develop materials by modifying one or two components before validation through experiments, there is no general complex material model for hybrid friction composite facing materials to correctly describe their behavior (e.g., in order to decrease testing costs).

The surface morphology of these materials lacks thorough investigation. There is no data available on specific wear coefficients when considering dry-friction facing material contacts with different surface roughness values or initial facing surface roughness.

The steady state and running conditions of a frictional life cycle are not well described in the literature.

The chemical reactions, physical changes, and effects of surface discontinuities are not yet clear, giving space for further optimization. Although TEI was thoroughly researched by scholars, the thermal test devices used typically investigate the final states, whereas the inner steps and mechanisms can only be extrapolated. Moreover, the effects on surface properties and morphology have not been deeply investigated. Friction and wear with surface changes are usually described by tests, with only a few finite element analyses reported from this field. The development of finite element software capabilities allowed the usage of complex models to describe friction, wear, or thermal aspects. These models were mainly developed to describe vibrations and thermal effects.

The aim of this paper was to illustrate the investigations necessary to determine the mechanical stiffness matrix parameters and thermal properties of a woven fiber-reinforced dry-friction material manufactured with given parameters. By revealing and solving challenges facing identification efforts, this study can be used to guide material identification methods for similar complex materials. The experimental results provide input parameters for thermomechanical simulation contact model development. The future aim is to cover friction material lifetime effects on dry clutch tribology by developing the model further.

## 2. Test and Model System

The effects of the tribological aspects of friction materials during their lifetime on transmission system properties can be modeled via a thermomechanical coupled contact model to create complex load cases. Thermomechanical models are usually used for investigations of transmissible torque loss due to heat peaks to provide comparable simulation data with certain test results. However, by widening the range of corresponding aspects considered, and by possibly decreasing the number of necessary tests, cost and time savings can be achieved. Moreover, these tests usually result in degradation of friction surfaces or permanent defects of the friction material, creating waste.

The model system illustrated in Figure 2 can be divided into two levels: thermomechanical (model level 1) and additional coupled tribological (tribomechanical) model (model level 2).

In addition to characterization of mechanical behavior, thermal properties for thermal responses are required. Depending on the dimensions of the model shown in Figure 2, the number of necessary parameters varies. With approximately axisymmetric geometries present in the clutch system, two-dimensional (2D) models are often utilized. On the other hand, during their lifetime, the surfaces of friction materials become unable to be specified through 2D simplifications. Therefore, properties describing three-dimensional behavior need to be determined.

For a thermomechanical coupled finite element contact model, as detailed in this study, a mechanical stiffness matrix is needed to set up the basic equation of a mechanical finite element method, i.e., the general Hook law.
(1)$$\sigma_{\mathrm{ij}}=C_{\mathrm{ijkl}}\cdot \varepsilon_{\mathrm{kl}}$$
where σ_ij_ is the stress tensor, C_ijkl_ is the stiffness matrix, and ε_kl_ is the strain tensor in a Descartes coordinate system. The stiffness matrix of orthotropic materials such as a clutch friction facing include nine independent parameters (1 = *x*; 2 = *y*; 3 = *z*)

(2)$$\left[\begin{array}{c} \sigma_{1}\\ \sigma_{2}\\ \sigma_{3}\\ \sigma_{4}\\ \sigma_{5}\\ \sigma_{6} \end{array}]=[\begin{array}{cccccc} C_{11} & C_{12} & C_{13} & 0 & 0 & 0\\ C_{21} & C_{22} & C_{23} & 0 & 0 & 0\\ C_{31} & C_{32} & C_{33} & 0 & 0 & 0\\ 0 & 0 & 0 & C_{44} & 0 & 0\\ 0 & 0 & 0 & 0 & C_{55} & 0\\ 0 & 0 & 0 & 0 & 0 & C_{66} \end{array}]\cdot [\begin{array}{c} \varepsilon_{1}\\ \varepsilon_{2}\\ \varepsilon_{3}\\ \varepsilon_{4}\\ \varepsilon_{5}\\ \varepsilon_{6} \end{array}\right]$$

Considering the created woven wreath structure of fiber reinforcement created during production, woven clutch facing friction materials are transverse orthotropic. Therefore, by organizing Equation (2), a flexibility matrix (inverse of stiffness matrix) is created.(3)$$\left[\begin{array}{c} \varepsilon_{1}\\ \varepsilon_{2}\\ \varepsilon_{3}\\ \varepsilon_{4}\\ \varepsilon_{5}\\ \varepsilon_{6} \end{array}]=[\begin{array}{cccccc} \frac{1}{E_{1}} & -\frac{\nu_{21}}{E_{1}} & -\frac{\nu_{31}}{E_{3}} & 0 & 0 & 0\\ -\frac{\nu_{12}}{E_{1}} & \frac{1}{E_{1}} & -\frac{\nu_{31}}{E_{3}} & 0 & 0 & 0\\ -\frac{\nu_{13}}{E_{1}} & -\frac{\nu_{13}}{E_{1}} & \frac{1}{E_{3}} & 0 & 0 & 0\\ 0 & 0 & 0 & \frac{1}{G_{23}} & 0 & 0\\ 0 & 0 & 0 & 0 & \frac{1}{G_{31}} & 0\\ 0 & 0 & 0 & 0 & 0 & \frac{2\left(1+\nu_{12}\right)}{E_{1}} \end{array}]\cdot [\begin{array}{c} \sigma_{1}\\ \sigma_{2}\\ \sigma_{3}\\ \sigma_{4}\\ \sigma_{5}\\ \sigma_{6} \end{array}\right]$$
where *E_i_* is the Young modulus, *G_i_* is the shear modulus, and *ν_i_* is the Poisson’s ratio.

To perform the tensile strength investigation of the cutout piece of clutch facing in terms of Young modulus and strength, DIN 53455 patent specifications were met on a 724002/2015 Zwick 10 kN experimental equipment with 5 mm/min pulling speed, as shown in Figure 3a. Measuring the initial surface area (A_0_) and length (l_0_) with the maximum force (F_max_), the Young modulus and strength values could be calculated using Equations (4) and (5).
(4)$$E=\frac{F_{\max}\cdot l_{0}}{A_{0}\cdot \Delta l}\;,$$
(5)$$\sigma_{B}=\frac{F_{\max}}{A_{0}}\;.$$

The two-directional strain measurement shown in Figure 3b and the Iosipescu shear test illustrated in Figure 3c are necessary to determine the elastic modulus, Poisson’s ratio, and shear modulus of the dry-friction clutch facing material.

The coefficients of thermal expansion in different directions, as well as the specific heat and thermal conductivity, were determined by experiments. The coefficient of thermal expansion describes how the size (for example length of volume) of an object changes with a change in temperature. To determine this value for the hybrid composite facing material, tests were conducted at the Thermophysical Property Investigation department (Applikationslabor Sektion Thermophysikalische Eigenschaften) of the laboratories of NETZSCH-Gerätebau GmbH in Germany using an N-5667-P-16 NETZSCH TMA 402 F1 Hyperion^®^ device. Figure 4 illustrates the elements of the apparatus on the left, equipped with a steel furnace capable of a temperature spectrum of −150 °C to 1000 °C. Here, 7–11 mm long test specimens were heated up to 180 °C at 5 K/min.

The automatic test execution, data acquisition, and evaluation were carried out using universal software for PC and control electronics. The software enables the determination of expansion coefficients, onset and peak temperatures, turning points, expansion speeds, density, etc.

Specific heat capacity can be described as the amount of energy that must be added, in the form of heat, to one unit of mass of a substance in order to cause an increase of one unit in its temperature. Differential scanning calorimetry (DSC) experiments were performed at the laboratory of Schaeffler Friction Products GmbH, Morbach to determine the specific heat.

The thermal conductivity coefficient is equal to the amount of heat conducted through the one unit of area of a substance due to one unit of temperature per unit of time. Specification of this value for the investigated friction material also took place at Schaeffler Friction Products GmbH Morbach.

Lee’s apparatus was utilized to determine the thermal conductivity coefficient, as illustrated in Figure 5. A disc (S) made of the investigated material was placed between two discs made of brass (B) and steel (M). Above them was placed a chamber (H) capable of steam conductivity. At steady state, T_1_ (T_A_) and T_2_ (T_B_) describe the temperatures of the steel and the brass discs, given at both sides of the investigated disc. Heat flow can be described by Equation (6), where the disc thickness is d and the cross-section is A.
(6)$$Q_{1}=\frac{\lambda \cdot A\cdot \left(T_{1}-T_{2}\right)}{d}$$

The heat loss of brass toward the environment can be described by Equation (7), where *c* is the specific heat capacity of brass and *dT/dt* is the cooling rate at T_2_.
(7)$$Q_{2}=m\cdot c\cdot \left(\frac{\mathrm{dT}}{\mathrm{dt}}\right)\cdot T_{2}\;.$$
(8)$$\lambda =\frac{m\cdot c\cdot \left(\frac{\mathrm{dT}}{\mathrm{dt}}\right)\cdot T_{2}}{A\cdot \left(T_{1}-T_{2}\right)}\;.$$

The necessary experiments for the mechanical and thermal identification of fiber-reinforced hybrid friction material of a dry-single plate clutch are summed up in Figure 6. Material identification test results then served as input parameters for a coupled thermomechanical finite element contact model that included effects of surface parameters during lifecycle and running under different operating conditions (initial surface roughness of the facing surface, sliding velocity, etc.) to provide a wider range of aspects covered by simulation, thereby cutting back testing costs.

## 3. Applied Composite Friction Materials

As illustrated in Figure 7, the first manufacturing steps for a conventional fiber-reinforced woven dry clutch facing involve wire preforming (A1: yarn manufacturing), dry mixing of fillers and modifiers (A2: compounding/granulating), and molding of mix around strand or wire preforms (B: coating). These are followed by the steps of a desired structure creation: weaving according to a specified pattern (C), then hot pressing (D), curing (E), and grinding (F). The recipe of such material components is often an industrial secret, making determination of the properties difficult through standard methods.

Components regarding the structure of the investigated hybrid composite friction facing of the dry sliding clutch disc can be divided into two main group of materials:long fiber reinforcement: yarn made of aromatic polyamide, glass fiber, polyacrylonitrile fiber, and copper,matrix (short fiber-reinforced composite with a hybrid indication), including a melamine-modified epoxy phenol resin with compounded rubber and other components such as sulfur, an aromatic polyamide as short fiber reinforcement, and other filler materials.

Figure 8 illustrates the results of an infrared (IR) spectral analysis conducted with the composite (due to short fiber reinforcement) “matrix” regarding its chemical composition. From the experiment, the following findings can be gathered:An –OH group is present according to the peak between 3000 and 3500 cm^−1^;Peaks below 3000 cm^−1^ refer to aliphatic C–H bonding;The peak at approximately 1500 cm^−1^ also refers to the above, in addition to other aliphatic-type bonds;No peaks refer to the presence of carbonyl or aromatic groups.

According to these assumptions, the investigated material was not polyester-based, but an epoxy-based resin.

## 4. Preparation of Test Samples

When preparing the test sample, the following factors were considered: test specimens with unchanged behavior and properties compared to the whole product they’re taken from are necessary for mechanical and thermal property identifications. Creating such pieces requires accurate and careful machining, controlled by a reliable system.

Our previous studies [23] found abrasive water jet machining to be an applicable technique for test sample creation, avoiding the heat generation effect of cutting methods and highlighting the effects of the presence of transfer medium (water) on the mechanical behavior of dry sliding clutch friction material. Figure 9 illustrates the cutting circumstances and the resulting surface.

Abrasive water jet machining is a nonconventional machining process where material is removed by impact erosion of high-pressure and high-velocity water and the entrained high velocity of grit abrasives on a work piece [24]. However, moisture of the environment should be avoided when utilizing a dry sliding clutch disc, due to the so-called cold judder phenomenon experienced, especially in humid circumstances, which definitely changes the frictional and mechanical parameters of clutch facings. Tensile tests were performed [23] according to DIN 53455, investigating three groups of specimens created by the 1515 MAXIEM OMAX abrasive water jet cutting machine from a clutch disc facing, and then they were sorted into three groups with different moisture levels. The first group of samples underwent a slight heat treatment (drying for 15 min, 150 °C), to mimic untouched specimens. Specimens of the second group were placed between moist layers for 15 min. The third group remained untreated in terms of moisture. As shown on Figure 10, the stress–displacement curves of three test samples with different moisture levels showed no significant difference, thus implying no harmful effects on the structure, whereby water jet cutting did not modify the material properties of the dry-friction clutch facing.

## 5. Results and Evaluation

### 5.1. Mechanical Material Model Creation Method

#### 5.1.1. Separation Idea and Rule of Mixtures

The mechanical properties of the material groups into which components were divided, as mentioned above, were determined separately, as detailed in a previous study [25], considering that components are industrial secrets and special orientations influence the production method. Since the utilization of rule of mixtures (ROM)−to create the mechanical stiffness matrix for the investigated friction material via the Tsai–Pagano equations [26] (see Equations (9)–(13), considering unidirectional composite materials or the Tsai–Pagano [27] model for randomly oriented fibrous composite—was described in this previous study, only the results are presented here.
(9)$$E_{11}^{*}=E_{m}{*V}_{m}+E_{f}{*V}_{f},$$
(10)$$E_{22}^{*}=\frac{E_{f}{*E}_{m}}{E_{f}-\sqrt{V_{f}}\left(E_{f}-E_{m}\right)},$$
(11)$$E=\frac{3}{8}E_{11}^{*}+\frac{5}{8}E_{22}^{*},$$
(12)$$G=\frac{1}{8}E_{11}^{*}+\frac{1}{4}E_{22}^{*},$$
(13)$$\nu =\frac{E}{2G}-1,$$
where E_m_, V_m_ are the elastic modulus and volume of the matrix component group, E_f_, V_f_ are the elastic modulus and volume of the long fiber reinforcement component group, E_11_* is the fictive elastic modulus in the direction of load, E_22_* is the fictive elastic modulus perpendicular to the direction of load, E is the effective elastic modulus, G is the effective shear modulus, and υ is the effective Poisson’s ratio.

These equations determine the fictive elastic modulus in the direction of load, as well as perpendicular to it, from the separately measured elastic modulus and volume of the matrix component and fiber reinforcement component groups, thereby obtaining the effective elastic modulus, effective shear modulus, and effective Poisson’s ratio of the composite.

#### 5.1.2. The Scientific Novelty of Utilizing Separation Idea and Rule of Mixtures for Model Creation

The utilization of separation before uniting gathered component material properties via the rule of mixtures is a common method for the property determination of fiber-reinforced polymer composite material. However, regarding the structure of materials, this method is usually applied to laminates. Its utilization for fiber-reinforced hybrid dry-friction clutch facing composites can be considered as a novel method. Furthermore, this novel utilization, as presented in this paper, can be applied in the following scenarios:when the production method gives a special orientation for the material,for example, for composites created via the scatter wound process or for composites with specially structured long fiber reinforcement component group,when conventional tests carried out on samples cut from the composite do not lead to relevant results, since the selection of sample contours within the product affects the results,when some ingredients are industrial secrets,when information regarding the contribution of different components to the strength of the whole structure is to be gained,for instance, when substitution is needed for one component or when immediate production support is required.

An example of the utilization of the calculation input data is detailed in Section 5.4, while setup, results and details of this simulation can be found in Appendix A.

#### 5.1.3. Properties of the Matrix Component Group

As illustrated in Figure 7, before yarn coating, the initial granulate components of the matrix went through an extrusion process to change their consistency for adhesion. The hypothesis was that the matrix of the composite was also a composite due to short fiber reinforcement.

In order to prove that these two groups of facings were created from the matrix material, the first group (group1) was made of granulate, hot-pressed and cured without extrusion. Creation of facings of the second group (group2) matched the whole facing production process, except for yarn coating. Tensile tests according to DIN 53455 were conducted to provide comparable data of mechanical behavior.

As shown in the stress–displacement graph (Figure 11), test samples cut from “matrix facings” created via the “extrusion” process (group 2) had twice the strength values of test samples lacking extrusion (group 1). In addition to these results, the presence of different short fiber reinforcements could be detected by merely tearing a piece of granulate. To conclude, the extrusion step in facing material production is essential to activate the strength-increasing effect of short fibers in the matrix.

Figure 12 illustrates the results of the above-mentioned tensile test on the matrix material test samples. Figure 13 shows the results of the two-directional strain measurement, while the Iosipescu shear test results are illustrated in Figure 14. The determined values of the elastic modulus, Poisson’s ratio, and shear modulus of the test specimens made of matrix material are listed in Table 2.

#### 5.1.4. Properties of the Long Fiber Reinforcement Component Group

For evaluating the mechanical properties of long fiber reinforcement yarn, standard tests of Schaeffler Savaria Ltd. were performed, creating braids from long fibers. The results are illustrated in Figure 15. Taking the fiber with the lowest Poisson’s ratio into account is a simplification toward safety.

From the measurements, the Young modulus of the long fiber reinforcement component group was found to be 27,300 MPa using the slope function in Excel (the vertical deviation divided by the horizontal deviation between any two points on the line, which is the rate of change along the regression line). The initial strain section was neglected due to a special hook shape supporting the braid for the test. A tighter wrapping led to an earlier increase in stresses. Braid 1 (blue) illustrates an example how fixing the braid on the hook can manipulate the results and behavior, forcing the evaluation to adapt to different strain sections. This group’s Poisson coefficient is represented by one component’s value, i.e., the glass fiber Poisson’s ratio of 0.2 [28]. The shear modulus was evaluated through calculation as 11,380 MPa.

#### 5.1.5. Creating the Mechanical Material Model

The calculation method for determining the material matrix components via the rule of mixtures was detailed in our previous studies [25]. The method hypothetically handles the material as if the reinforcement was parallel to the loads and creates stiffness matrix elements via the Tsai–Pagano equations with the utilization of the rule of mixtures for randomly oriented composites (see Equations (14)–(19)) for the two hypothetical laminas with an orientation of 90°–47° to the circumferential direction.
(14)$$E_{11}=E_{m}*V_{m}+E_{f}*V_{f}=14.3\;GPa,$$
(15)$$E_{22}=\frac{E_{f}*E_{m}}{E_{m}*V_{f}+E_{f}*V_{m}}=6.78\;GPa,$$
(16)$$\upsilon_{12}=\upsilon_{m}*V_{m}+\upsilon_{f}*V_{f}=0.3,\;$$
(17)$$\upsilon_{21}=\frac{E_{22}}{E_{11}}*\upsilon_{12}=0.144,\;$$
(18)$$G_{12}=\frac{G_{f}*G_{m}}{G_{m}*V_{f}+G_{f}*V_{m}}=2.1\;GPa,$$
(19)$$C^{-1}=S\;=\left[\begin{array}{ccc} \frac{1}{E_{11}} & -\frac{\upsilon_{12}}{E_{11}} & 0\\ -\frac{\upsilon_{21}}{E_{22}} & \frac{1}{E_{22}} & 0\\ 0 & 0 & \frac{1}{G_{12}} \end{array}\right]=\left[\begin{array}{ccc} \frac{1}{14.3} & -\frac{0.3}{14.3} & 0\\ -\frac{0.11}{6.78} & \frac{1}{6.78} & 0\\ 0 & 0 & \frac{1}{2.1} \end{array}\right]\mathrm{GPa}$$

Then, by considering a cylindrical coordinate system and the fact that the fibers are positioned at 47° and −47° (λ = 47°) to the radial coordinate direction, a transformation matrix detailed in Equation (20) was set up to transform the results according to Equation (21) so that the investigated material could be handled as a quasi-laminate in a cylindrical coordinate system [r,φ,z], and the results are presented in Table 3.
(20)$$T=\left[\begin{array}{ccc} cos^{2}\lambda & sin^{2}\lambda & 2sin\lambda cos\lambda \\ sin^{2}\lambda & cos^{2}\lambda & -2sin\lambda cos\lambda \\ -sin\lambda cos\lambda & sin\lambda cos\lambda & cos^{2}\lambda -sin^{2}\lambda \end{array}\right],$$
(21)$$S^{\prime }=T^{t}\cdot S\cdot T.$$

An extensional stiffness matrix illustrated by Equation (22) was created to unite the two identical hypothetical laminas, resulting in the determination of the quasi-laminate composite facing material properties listed in Table 4.
(22)$$A=\overset{n}{\underset{i=1}{\sum }}{C^{\prime }}_{i}*h_{i},$$
where *C′_i_* is the stiffness matrix of lamina *i*, *h_i_* is the thickness of lamina *i* (1.8 mm in our case), and *n* is the number of laminas.

A few limitations should be noted for the utilization of this model. This material model fails to take the changes in mechanical properties due to temperature into consideration, whereas it also lacks the effects of fiber–fiber and fiber–matrix adhesion on the elements of the stiffness tensor. The former can be explored through thermal aspects, while quantifying adhesion in such a material system via traditional methods has its difficulties, since the investigated matrix is not thermoplastic and vulcanization of fibers without coating results in a 70% loss in strength (measured by aromatic polyamide). Therefore, tests like the micro-droplet method using a single fiber cannot be performed. However tensile tests performed on single-fiber-reinforced samples created via a unique production line method revealed that the weak link of fiber–fiber and fiber–matrix adhesion is the glass fiber–glass fiber adhesion. The fact that molecular interactions are also affected by the thermal expansion of the matrix, the surface roughness, and different inner and outer forces makes it difficult to include these elements. Moreover, the effect of heat treatment on fiber properties is neglected in the ROM method, whereas it is considered when measuring the thermal properties of specimens cut out from facings after a complete production procedure including heat treatment.

### 5.2. Thermal Aspects

#### 5.2.1. Coefficient of Thermal Expansion

Figure 16, Figure 17 and Figure 18 illustrate the coefficient of thermal expansion (CTE) experimental results versus temperature in different directions (radial, axial, and normal, i.e., parallel to thickness) regarding the test sample geometry. Investigations were performed in air from 0 °C to 180 °C (heating speed 5 K/min) on 7–11 mm long test samples.

During operational conditions, the range of temperature rises to 300–500 °C [29] at certain contact points on the facing. On the other hand, considering the whole facing, the friction facing is heated to approximately 200 °C. Therefore, measurements were performed up to 180 °C with two heat-up loops.

Figure 16 shows CTE measurement results in the radial direction. Both heat-up curves show an initial transient phenomenon up to 50 °C before an increasing trend is presented. The differences in the length of the initial transient section and the steepness above 60 °C are related to the different effects of heating and the high temperature (at the end of the first heat-up loop) on different yarn CTE values. In the tangential direction illustrated in Figure 17, after the initial section, the composite friction material experiences a slow decline; then, a rapid drop occurs before reaching the minimum values. After the heat-up loop, this decline is not present. Figure 18 shows the results in the normal direction. In this direction, behavior is mainly governed by hybrid matrix characteristics; therefore, no significant difference can be detected between the two heat-up curves, suggesting that the fiber structure is more sensitive to residual heat-induced stresses.

In addition to the average expansion coefficient in the radial direction, an irreversible 0.09% decrease in the length of the sample was measured (not illustrated here) between the first and the second heating, as also detected in the two other directions. The highest values were measured in the normal direction.

#### 5.2.2. Specific Heat Capacity

Considering the above-mentioned operational temperature conditions, the specific heat versus temperature graph showed a stable behavior.

Figure 19 shows the specific heat capacity of the investigated facing material versus temperature.

#### 5.2.3. Thermal Conductivity Coefficient

The thermal conductivity measurements were conducted using Lee’s apparatus. According to the measurement results detailed in Table 5, the thermal conductivity coefficient of the investigated clutch facing was 0.398 W/(m∙K).

### 5.3. Modeling

Current simulation methods of automotive dry clutches usually utilize 2D axisymmetric geometry models, not only to reduce calculation time in a coupled thermo-mechanical finite element model, but also due to the fact that the 3D behavior of the clutch facing is only well described for axial loads. Current models usually lack the utilization of loads or lifetime-induced variation in mechanical, thermal, or tribological parameters responsible for contact relations. Modeling such deviations in 2D is almost impossible. Despite burst speed being one of the most dangerous load cases of the clutch facing, 2D investigations lack the potential to consider such loads or how the part is affected in that aspect. Comparing 2D and 3D coupled thermomechanical models of the transmission system of an automotive clutch, the solution time may increase upon taking the problem out of one plane to 3D, but the model set-up considerably decreases, since there is no need to validate 2D and 3D stiffness and their relationship after geometric simplifications of different parts for correct results.

This paper introduced the necessary parameters to be determined by performing measurements for the complete mechanical and thermal description of a dry sliding hybrid polymer composite material.

By utilizing ROM and conducting tensile tests, Iosipescu shear tests, and two-directional strain measurements, parameters describing mechanical material behavior via a stiffness matrix were obtained. By performing a thermal expansion test and differential scanning calorimetry, as well as utilizing Lee’s apparatus, the characterization of thermal aspects was achieved.

The parameters investigated and defined in this paper were provided as inputs for a finite element thermomechanical contact model of a dry automotive clutch, as illustrated in Figure 20. With the method described here, the effect of new components or completely new materials could be considered. Finite element software (Version 18.2, Ansys, Inc., Canonsburg, WA, USA) covers the inclusion of orthotopic and temperature-dependent material behavior with user-defined material models and coupling of load cases.

The aim of this model in the future is to widen simulation possibilities through tribological aspects (such as differentiating friction and wear conditions under different loads or mileage), regarding mechanical, thermal, and tribological performance verification tests of the friction facing aiming at cost reduction, while identification investigations can provide methods for newly created material parametrization.

### 5.4. Simulation in Finite Element Software Environment

A mechanical finite element model was set up in order to illustrate the behavior of the facing material with the calculated mechanical properties under burst speed load. ANYS Workbench (Version 18.2, Ansys, Inc., Canonsburg, WA, USA) served as the simulation environment for the calculation. Figure 21 illustrates the material property input parameters. The calculation set-up and results are detailed in Appendix A.

## 6. Conclusions

A remarkable development defines the history of automotive clutch friction facing materials, during which construction and material discoveries and efforts resulted in fiber-reinforced hybrid composites becoming the most commonly used dry-friction materials in the industry.

Novel materials open novel fields of investigations, not only in terms of identification, but also in terms of challenges for researches to handle during sample creation before conducting any tests or completing characterization. With potential damage being a factor, abrasive water jet machining turned out to be an efficient application for the creation of test samples for mechanical and tribological investigations of hybrid composite dry-friction clutch facing materials.

Modeling the complex hybrid mechanical material system of a woven fiber yarn (glass fiber with aromatic polyamide, copper, and PAN) reinforced friction material was performed with the idea of creating separated component groups (matrix and fiber groups) and ROM.

IR spectral analysis conducted with the composite (due to short fiber reinforcement) “matrix” regarding its chemical composition revealed that the matrix itself is an epoxy-based composite.

Furthermore, tensile test investigations of the matrix of the composite, whose components are industrial secrets, revealed the presence of suspected short fiber reinforcement and its effects on material strength of the hybrid composite material after the extrusion production step.

From further tensile tests, Iosipescu shear tests, and two-directional strain measurements, the values mechanically describing the matrix component group were as follows: Young modulus 4290 MPa; Poisson’s ratio 0.38; shear modulus 1290 MPa.

The fiber reinforcement component group was characterized by the following values: Young modulus 27,300 MPa; Poisson’s ratio of glass fiber 0.2; shear modulus 11,380 MPa. Tensile strength investigations also revealed that the weak link of fiber–fiber and fiber–matrix adhesion is the glass fiber–glass fiber adhesion.

Component group values were united via ROM, resulting in a mechanical stiffness matrix for the whole dry-friction fiber-reinforced hybrid composite material.

Thermal investigations defined, in three geometrically specified directions, the coefficient of thermal expansion between 0 °C and 180 °C, the specific heat versus temperature between 50 °C and 240 °C, and the thermal conductivity coefficient as 0.398 W/(m∙K).

The scientific novelty of the model was granted by a method determining the stiffness matrix and its components, along with the results of thermal investigations. These properties, together with the method itself, contribute to a thermomechanical model that allows characterization of fiber-reinforced woven hybrid friction composites with given production parameters from almost raw material properties of fiber-reinforced hybrid composite dry-friction clutch facings.

Moreover, this study can act as an effective reference with guidance for material identification methods for similar complex materials, especially when the production method gives a special orientation for the material, when the selection of sample contours within the product would affect the results of material investigations, when some ingredients industrial secrets, or when information about the contribution of different components to the strength of the whole structure is to be gained.

Future investigations will reveal the tribological behavior and its effects during the component’s lifetime to extend the prediction capability of currently utilized contact models by creating a coupled thermomechanical and tribomechanical model.

## Figures and Tables

**Figure 1 materials-13-04508-f001:**
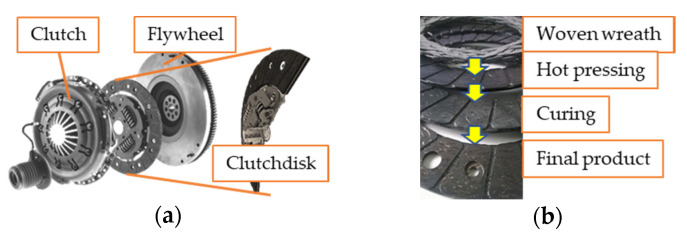
(**a**) Parts of a dry clutch system and cutout image of the dry-friction clutch disc; (**b**) products of phases of dry-friction clutch material production.

**Figure 2 materials-13-04508-f002:**
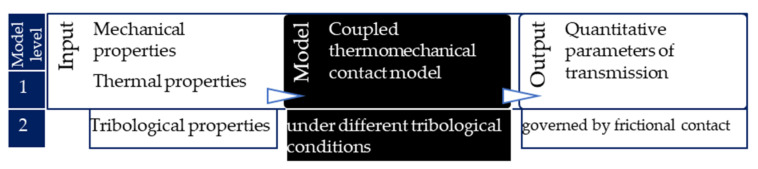
Schematic figure of the model for which input parameters are to be identified.

**Figure 3 materials-13-04508-f003:**
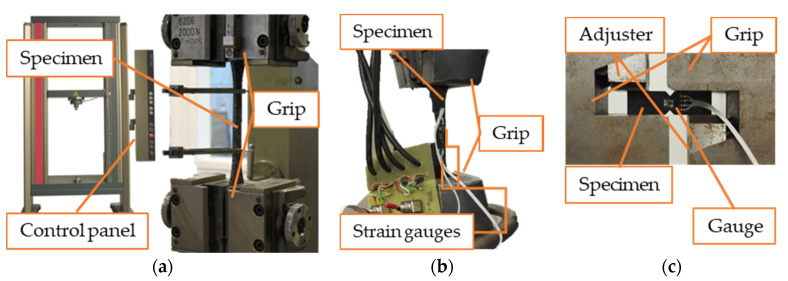
(**a**) Zwick 10 kN experimental equipment; (**b**) two-directional strain measurement set-up; (**c**) Iosipescu shear test.

**Figure 4 materials-13-04508-f004:**
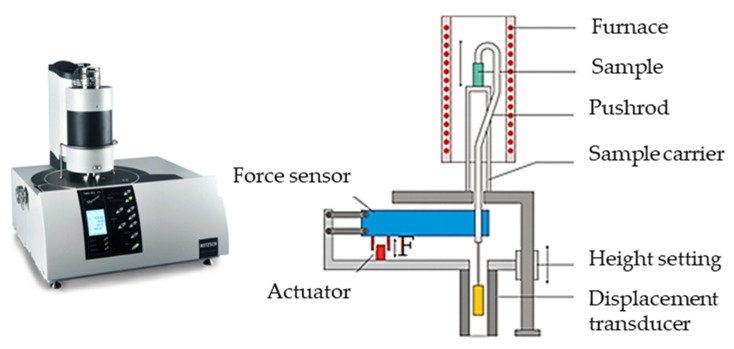
NETZSCH TMA 402 F1 Hyperion^®^ device and its elements used for the measurements of thermal expansion coefficient.

**Figure 5 materials-13-04508-f005:**
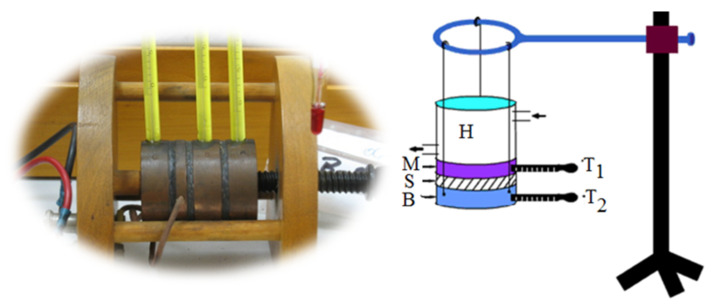
Lee’s apparatus.

**Figure 6 materials-13-04508-f006:**
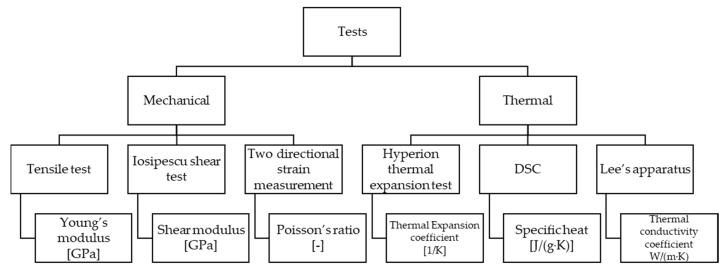
Material identification test summarization.

**Figure 7 materials-13-04508-f007:**
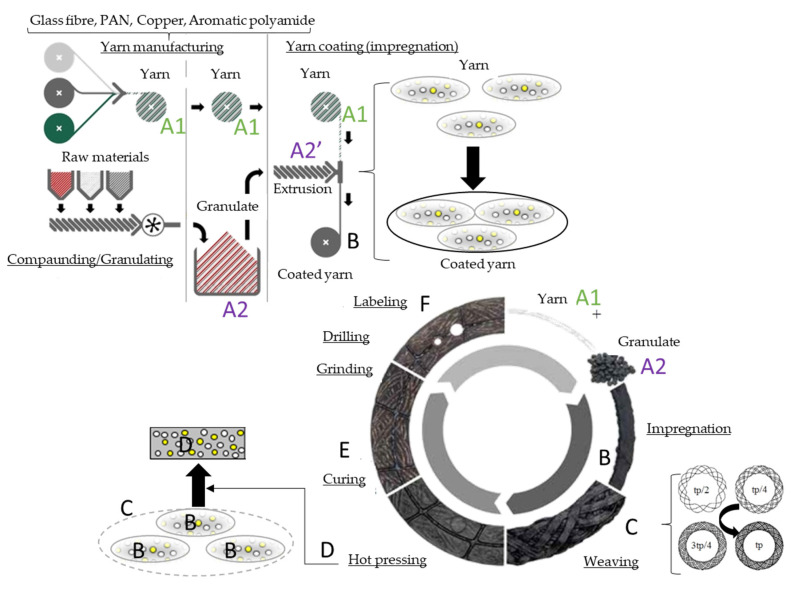
Production steps of the hybrid composite conventional dry-friction clutch facing.

**Figure 8 materials-13-04508-f008:**
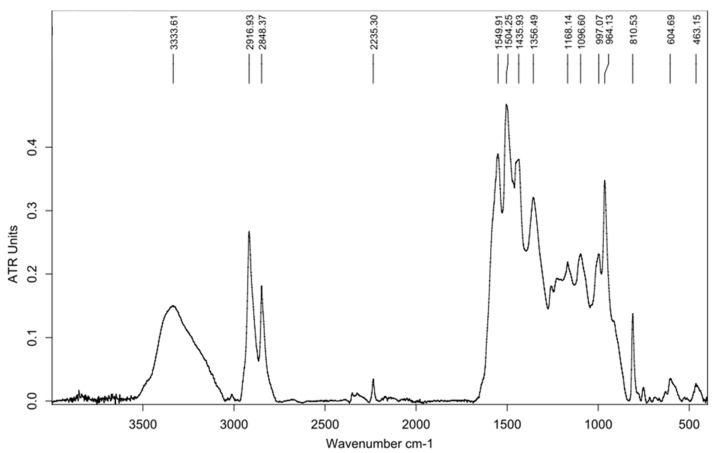
IR spectral analysis of the chemical composition of the matrix of the hybrid composite.

**Figure 9 materials-13-04508-f009:**
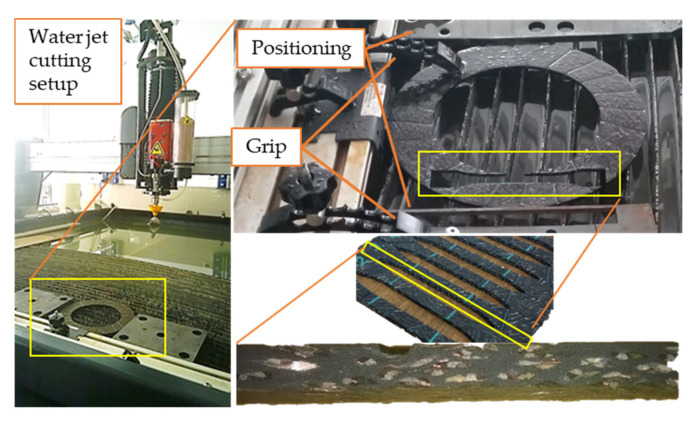
The 1515 MAXIEM OMAX abrasive water jet cutting machine and the smooth cutting edge of a test specimen.

**Figure 10 materials-13-04508-f010:**
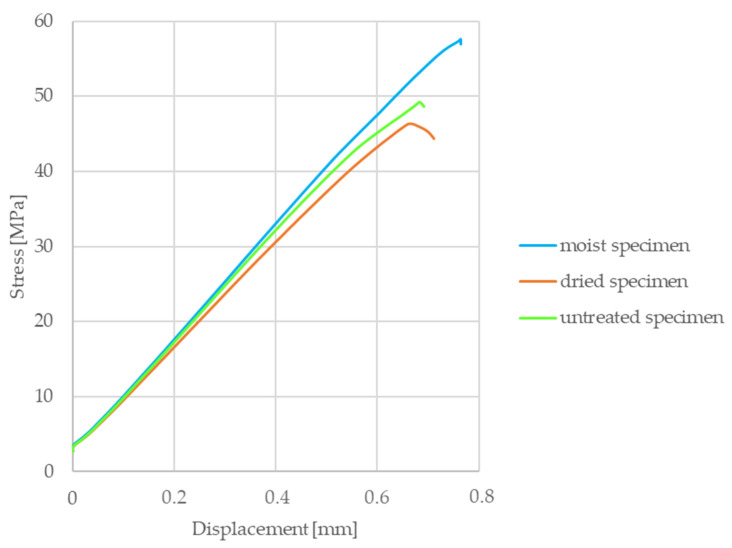
Force–displacement curves of test samples with different moisture levels.

**Figure 11 materials-13-04508-f011:**
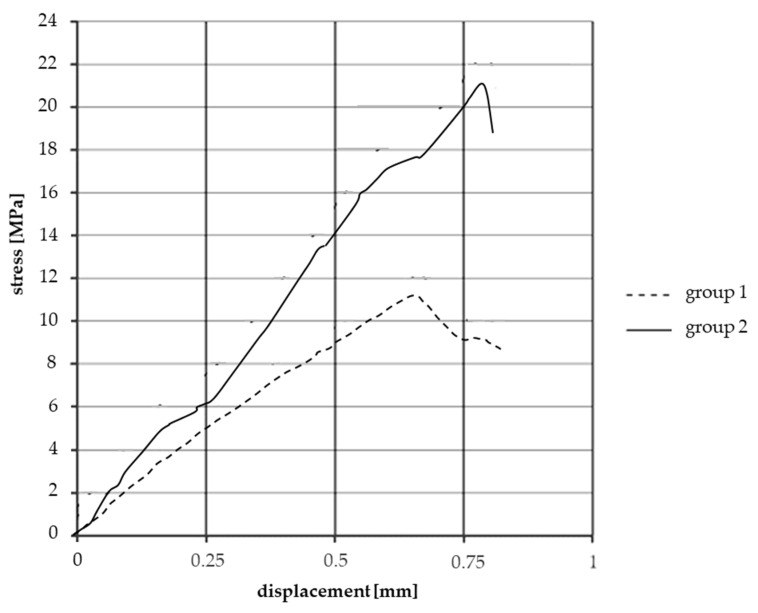
Effects of extrusion on the strength of test samples from the matrix component group.

**Figure 12 materials-13-04508-f012:**
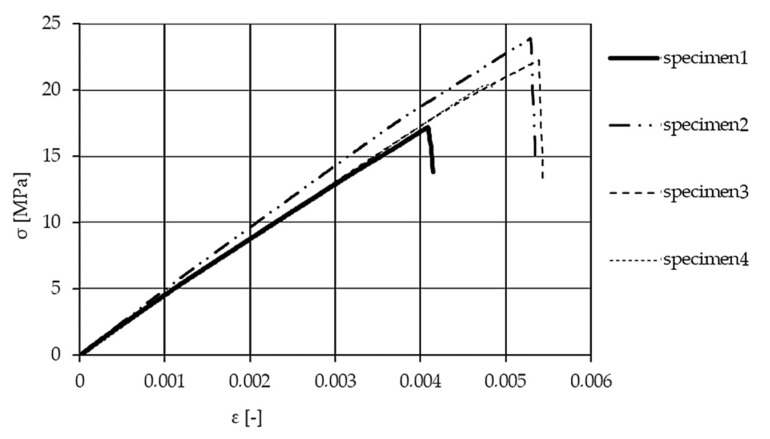
Tensile test results: stress–strain curve of the matrix component group.

**Figure 13 materials-13-04508-f013:**
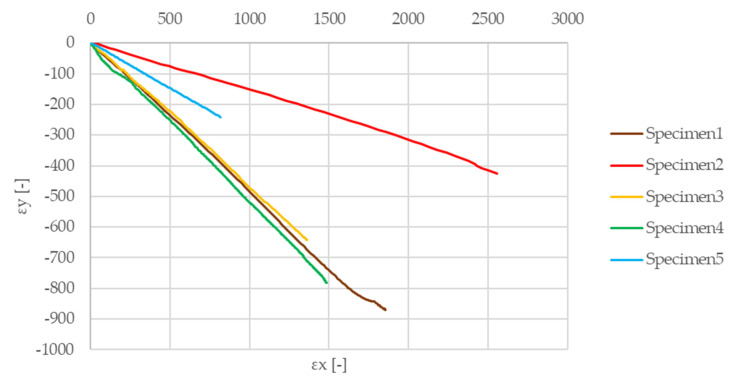
Poisson coefficient measurement via two directional strain-detection.

**Figure 14 materials-13-04508-f014:**
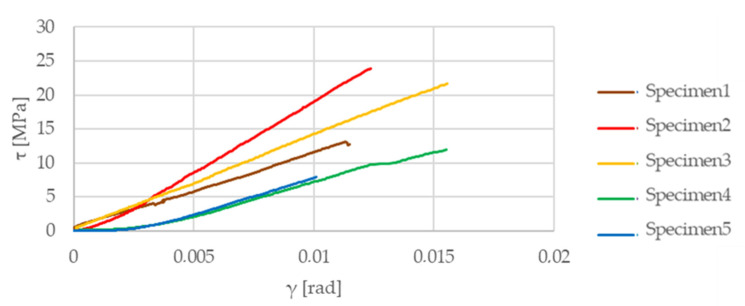
Iosipescu shear test results of specimens.

**Figure 15 materials-13-04508-f015:**
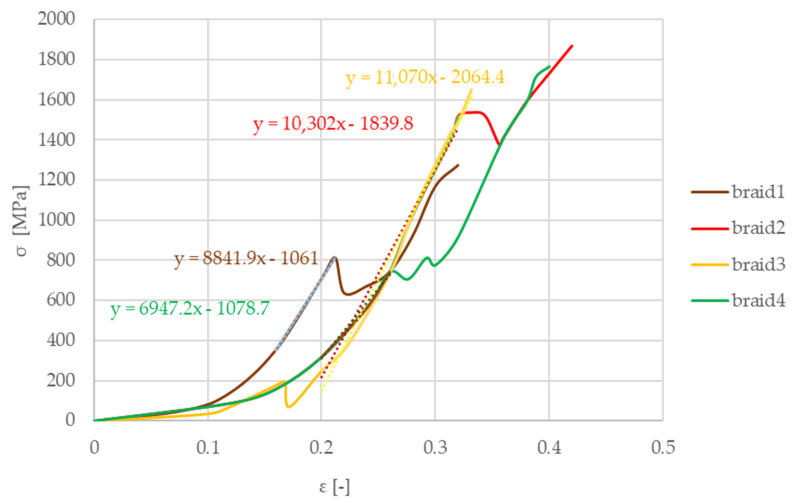
Stress–strain curves of woven braids (yarn) from the long fiber reinforcement group.

**Figure 16 materials-13-04508-f016:**
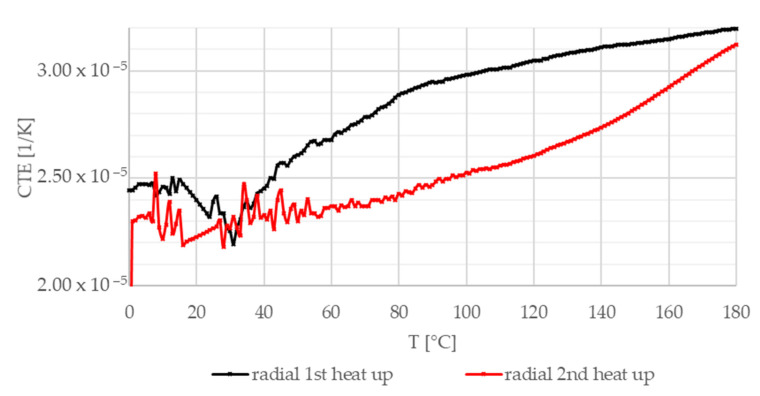
Coefficient of thermal expansion—radial direction, two heat-up loops (black, red).

**Figure 17 materials-13-04508-f017:**
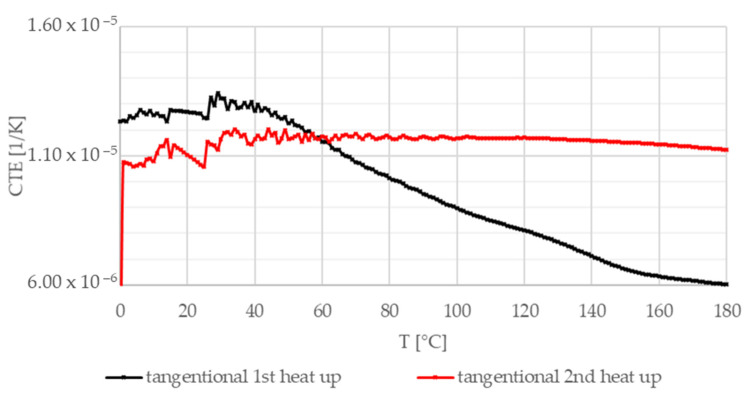
Coefficient of thermal expansion—tangential direction, two heat-up loops (black, red).

**Figure 18 materials-13-04508-f018:**
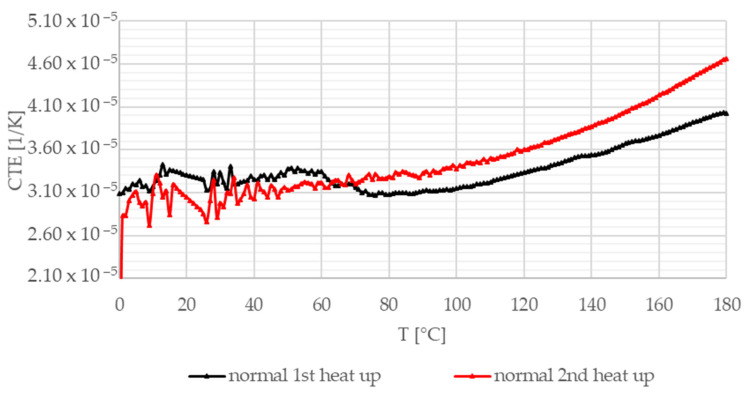
Coefficient of thermal expansion—normal direction (thickness), two heat-up loops (black, red).

**Figure 19 materials-13-04508-f019:**
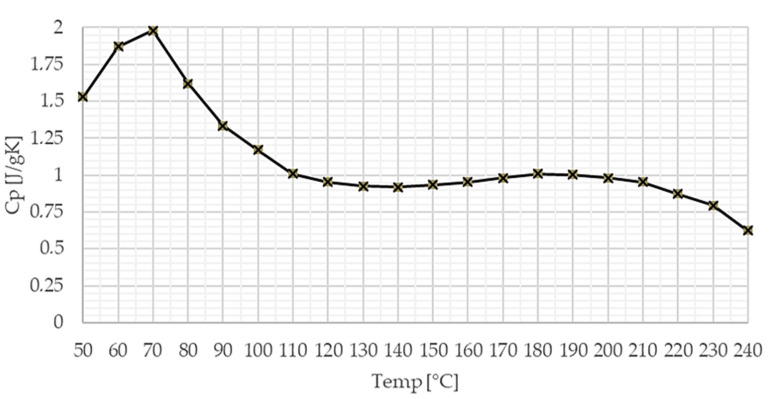
Specific heat capacity of the investigated facing material versus temperature.

**Figure 20 materials-13-04508-f020:**
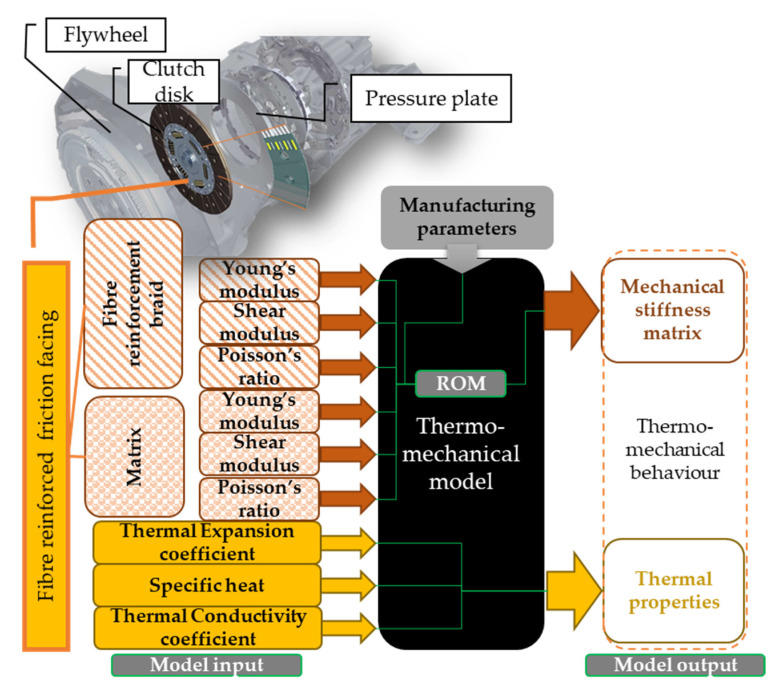
Schematic representation of a coupled thermomechanical model with geometries of a dry automotive clutch contact and necessary input parameters and capabilities to describe the behavior of a fiber-reinforced hybrid composite facing under loads.

**Figure 21 materials-13-04508-f021:**
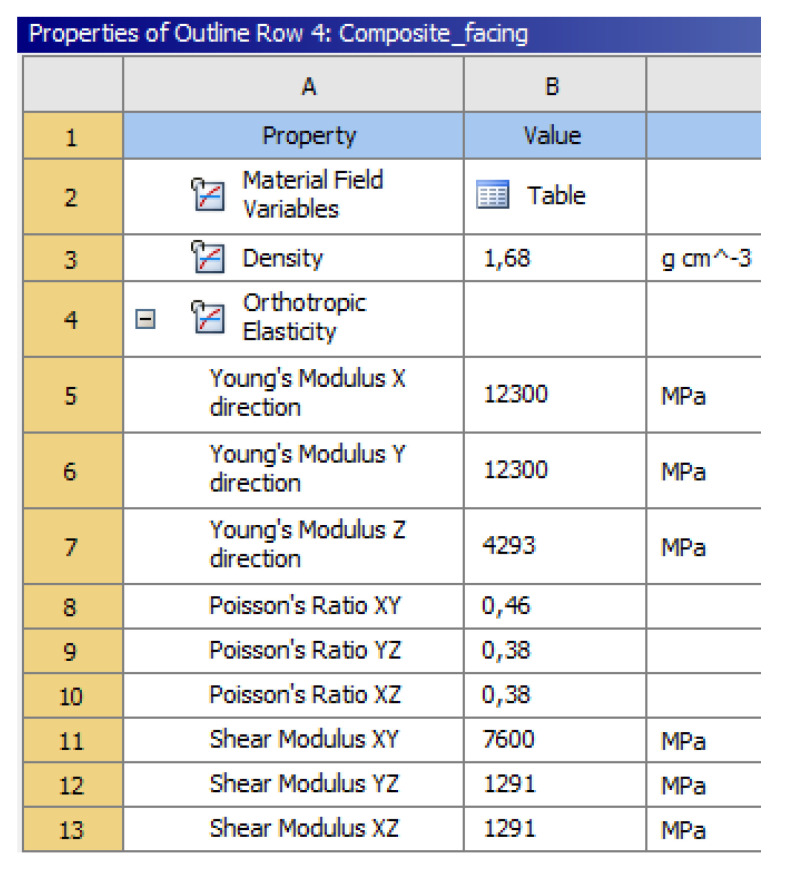
Input mechanical material property parameters in ANSYS environment.

**Table 1 materials-13-04508-t001:** Clutch friction materials vs. constructions (adapted from [2]).

	Year	1886	1889–1920s	1918–1920s	1920s	1900–1918	1900s	1920s
	Type of Clutch	Transmission Belt Clutch	Friction Clutch					
Year	Friction Material		Cone/Bevel Friction Clutch	NAG Cone	Daimler Al-cone	Daimler/Mercedes Spring Band Clutch	Weston Multidisc Oiled/Dry	Single-Disc Clutch
1886	Leather belts	x						
1889	Camel hair		x	x				
After 1889	Leather belt soaked into castor oil		x					
After 1889	Spring-loaded pins/ leaf spring + leather		x					
1918	Metal		x			x		
1918	Oiled aluminum				x			
1900s	Oiled bronze and steel						x	
1900s	Riveted friction lining						x	
1920s	Graphite lubricated							x
1920s	Ferodo asbestos							x
1990s	Asbestos-free linings							x

**Table 2 materials-13-04508-t002:** Properties of the matrix component group.

**Young Modulus (MPa)**	4290
**Poisson’s Ratio**	0.38
**Shear Modulus (MPa)**	1290

**Table 3 materials-13-04508-t003:** Properties of the fiber-reinforced composite in the cylindrical coordinate system.

Facing Mechanical Properties in Cylindrical Coordinate System	
E_rr_ (GPa)	6.06
E_φφ_ (GPa)	6.26
ν_rφ_	0.454
ν_φr_	0.469
G_rφ_ (GPa)	3.83

**Table 4 materials-13-04508-t004:** Properties of the fiber-reinforced composite as a quasi-laminate.

Facing Mechanical Properties via Extensional Stiffness Matrix	
E_11_ (GPa)	12.3
E_22_ (GPa)	12.3
E_33_ (GPa)	4.3
ν_12_	0.46
ν_23_	0.38
ν_13_	0.38
G_12_ (GPa)	7.6
G_23_ (GPa)	1.3
G_13_ (GPa)	1.3

**Table 5 materials-13-04508-t005:** Thermal conductivity coefficient investigation results.

Voltage U	6.3	6.36	6.37	6.4	6.53
Current I	0.296	0.296	0.296	0.298	0.296
t_0_ room temperature	25.1	20.8	22	22.8	25.4
t_A’_	40.8	38.7	38.4	39	41.8
t_A_ (t_A’_–t_0_)	15.7	17.9	16.4	16.2	16.4
t_B’_	44.6	42.4	42.2	42.8	45.4
t_B_ (t_B’_–t_0_)	19.5	21.6	20.2	20	20
t_C’_	45.9	43.5	43.4	44.2	45.8
t_C_ (t_C’_–t_0_)	20.8	22.7	21.4	21.4	20.4
t_B_–t_A_	3.8	3.7	3.8	3.8	3.6
e	0.0013	0.0012	0.0012	0.0013	0.0013
λ [W/(m∙K)]	0.38	0.43	0.387	0.39	0.428

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
