# Peer review of "Micromechanical Model and Thermal Properties of Dry-Friction Hybrid Polymer Composite Clutch Facings"

_materials, 2020, doi:10.3390/ma13204508_

Round 1
Reviewer 1 Report
Dear Authors,
In your work you describe the methods for assessing parameters for simulation of thermomechanical problem. You present the obtained results but you do not apply them to any simulation. You article looks like a chapter of a book. Your paper is an excellent guide for anyone who want to prepare model for numerical simulation of thermomechanical problem however it is not proper for publication as original scientific paper. Some editorial problems you can find in attached file.
Best regards

Reviewer 2 Report
This is an educational paper and can be used as a good guideline to obtain material properties needed for finite element analysis, but it lacks novelty. In this work, the mechanical properties of matrix and fibers were measured separately, and then the rule of mixture was used to find mechanical properties of the composite while testing the composite to obtain the material properties provides more accurate results. The reason provided by the authors for choosing this approach is not convincing. Moreover, the authors also did not choose the same approach for testing the thermal properties of the composite. In the end, I expected to see at least a simple finite element analysis using these parameters.
Following please find my comments to improve the manuscript:
Some of the sentences are too long and difficult to follow and need to be revised.
Good schematics have been used. The colors used in table 1 and figure 2 can be improved to enhance the readability.
Please use labels for different components of experimental equipment in figure 3. Figure 3.b is not illustrative enough.
The labels used for Figure 11 are not explanatory, you can use “group 1” or “cured without extrusion” instead of gr2 and “group 2” or “cured without extrusion” or “complete process without yarn coating” instead of “vhk1”.
In Figure 13, the orders of strain values are wrong. If you scaled up the values, you must mention it in the manuscript.
In Figure 15, you should use an arrow to indicate which equation corresponds to which section of the curve.
In Section 5.1.3. you need to explain what are braid 1 to 4.
In Section 5.1.4 you must show the equations you used for your calculations.
In Table 4, you need to correct the superscripts.
Reviewer 3 Report
Comments
This paper investigated hybrid polymer composites. The outcome is interesting for readers. However, there are several aspects that need to be improved. The reviewer can only recommend for publication if the author satisfactorily address the following comments in the revised version.
- Figure 9 is very unclear and suggest to provide a clear picture for specimen.
- Figure 11: the initial load at zero mm displacement can be removed from the graph.
- How about impact and fatigue behavior of hybrid polymer composites? The literature section can be improved by highlighting the performance of hybrid composites under impact [Ref: A novel hybridised composite sandwich core with glass, kevlar and zylon fibres–investigation under low-velocity impact] and fatigue [Ref: Testing and modelling the fatigue behaviour of GFRP composites–Effect of stress level, stress concentration and frequency] loading scenarios.
Round 2
Reviewer 1 Report
Dear Author,
Thank you again for choosing "Materials" for publication of Your paper. After Your improvements (especially Appendix A) I can recommend it for publication. Nevertheless I have some minor hints that are marked in attached file.
Best regards

Reviewer 2 Report
After the improvements made to the manuscript, I recommend it for publication.
